# Stochasticity and Drug Effects in Dynamical Model for Cancer Stem Cells

**DOI:** 10.3390/cancers15030677

**Published:** 2023-01-21

**Authors:** Ludovico Mori, Martine Ben Amar

**Affiliations:** 1Laboratoire de Physique de l’Ecole Normale Supérieure, Ecole Normale Supérieure, Université PSL, CNRS, 75005 Paris, France; 2Institut Universitaire de Cancérologie, Faculté de Médecine, Sorbonne Université, 91 Bd de l’Hôpital, 75013 Paris, France

**Keywords:** cancer stem cells, tumor heterogeneity, stochasticity, plasticity, activator, inhibitor, cytotoxic and cystostatic drugs, ultrasounds

## Abstract

**Simple Summary:**

Phenotypical heterogeneity constitutes a feature of tumors that strongly impacts their growth and stability as well as possible therapies. Among the biological models that take this aspect into account we considered the Cancer Stem Model, which assumes the tumor population to be composed of stem cells and differentiated cells. Aim of our study was to include biologically originated stochastic factors in the model and investigate its impact on the tumor evolution numerically. In addition, we developed a model compatible with our main system in order to describe possible therapies, and considered their outcomes in some examples. Our results are consistent with state-of-the-art research in the field and confirm the descriptive power of the Cancer Stem Model.

**Abstract:**

The Cancer Stem Model allows for a dynamical description of cancer colonies which accounts for the existence of different families of cells, namely stem cells, highly proliferating and quasi-immortal, and differentiated cells, both undergoing cellular processes under numerous activated pathways. In the present work, we investigate a dynamical model numerically, as a system of coupled differential equations, and include a plasticity mechanism, of differentiated cells turning into a stem state if the stem concentration drops low. We are particularly interested in the stability of the model once we introduce stochastically evolving parameters, associated with environmental and cellular intrinsic variabilities, as well as the response of the model after introducing a drug therapy. As long as we stay within the characteristic time scale of the system, defined on the base of the needed time for the trajectories to converge on stable states, we observe that the system remains stable for the main parameters evolving stochastically according to white noise. As for the drug treatments, we discuss a model both for the kinetics and the dynamics of the substance in the organism, and then consider the impact of different types of therapies in a few particular examples, outlining some interesting mechanisms, such as the tumor growth paradox, that possibly impact the outcome of therapy significantly.

## 1. Introduction

Solid tumors, primary or metastatic, are heterogeneous at all stages of the patient’s lifespan: they can contain any kind of cell present in the host organ and attract different cells of the immune system such as macrophages, fibroblasts and lymphocytes inside the organ or in the stroma. Focusing only on cancerous cells, different categories have been identified, which mostly depend on their functions in the tumor. On top of the hierarchy, there are cancer stem cells, a small group of cells that are considered tumor initiators, being very proliferative and resistant to classical oncologic treatments. They are called stem since, similarly to standard stem cells in embryology, they can divide symmetrically or asymmetrically into a cascade of progenitors, as well as numerous intermediate cells, before creating fully differentiated cancerous cells. The cancerous cell assembly alone is very heterogeneous in composition and spatially inhomogeneous, which is among the reasons for the failure of therapeutics. For biologists and clinicians, the Cancer Stem Model (CSM) [1] explains among other things the relapse of tumors after full treatment in situ, resistance to radio and chemotherapies, and the mechanisms of EMT transition and metastasis. The aim of this paper is to transpose mathematical and physical concepts to this biological model and to explain and eventually predict some aspects of cancer evolution. As often in this area, simplifications are necessary if one is interested in understanding the main lines of evolution of this pathology. These reductions of traits are often misunderstood by the biologists who, after several decades, succeeded in identifying the correct biomarkers [2] according to the tumor localization in the body [3] and the main key pathways responsible for the stemness of these specific cells [4], such as Nanog,Wnt/β catenin, hedgehog, Notch, Jak/STAT and P13K/AKt. Some are associated with tumor initiation, others with innate resistance, whereas all of them appear to be at the origin of self-renewal differentiation.

In the past decades, biophysicists have used theoretical models from dynamical systems to effectively describe the competition in a living community at all scales, ranging from predator-prey models in conditioned environments, with the notable example of Lotka-Volterra equations, to cell population dynamics [5,6]. This approach may represent faithfully the time-dependent coupling between the main actors of the community, such as the different kinds of cells and the different pathways clustered under their role as activators (for instance MEK, mitogen-activated kinase), inhibitors, or responsible for de-differentiation (TGF-β). Along with these developments, the idea that unpredictable environmental factors could play a significant role in otherwise deterministic models started gaining some recognition. If the underlying ecological mechanisms were previously based on intuition, in more recent decades developments in population models led to deductive principles accounting for the role of the environment [7]. In many cases, as for computational oncology, one makes strong assumptions by reducing multiple coupled interactions to one parameter of the system. Being subject to fluctuations induced by internal processes, and competition between individuals and the environment, this parameter, therefore, varies stochastically [8,9,10]. This mechanism embodies the internal variability of the system as well as the ways it is affected by a number of intrinsically mutable environmental quantities [11]. Stochasticity becomes thus an ingredient in dynamical investigations of population models aimed at testing the model under more realistic conditions and exploring possible new and at times astonishing behaviors [12,13]. We are here considering a simplified dynamical model of a heterogeneous cancer cell population which is well suited if we want to study the impact of stochasticity. In particular, we want to know whether and under what conditions the Cancer Stem Model is stable and therefore finds confirmation after the introduction of stochastically fluctuating factors and parameters.

Models for cancer cell populations are often conceived with special attention to the potential clinical implications [14,15,16], and particularly the development of therapies. A description of drugs using dynamical systems is suited for a quantitative approach over pharmacology, which requires a cancer population model compatible with the various therapeutic possibilities [16,17,18]. The aim of this work, in addition to investigating environmental stochasticity, is to describe and include drug treatments in a model of cancer stem cells. We will explore the possibilities of therapeutic treatments by developing a model for the kinetics and the dynamics of the drug, and extending therewith the previously employed cancer population model. It will be possible to compare, by looking at some specific examples, the possible outcomes of different types of treatments, and to draw some conclusions regarding their efficacy.

## 2. The Heterogeneity of Tumors Explained by the CSC Model

We consider the population to be formed by 3 different types of cells, which include: cancer stem cells (CSCs), a small sub-population that exhibits a high proliferation rate and quasi-immortality [19]; differentiated cancer cells (DCs), that consist of tumor-specific cells that vary between organs and have specific properties; and other cells that do not contribute significantly to the dynamics of the tumor, such as dead, quiescent and healthy cells. In this modeling, we skip the intermediate steps concerning the progenitors.

The CSCs are the only family in the model able to undergo cell division and are thus responsible for the proliferation of the colony. Their reproduction channels are regulated by chemical activators/inhibitors and growth factors generated by the cells themselves or the stroma. These factors are the other dynamical variables in the model in addition to the different kinds of cells. One CSC can either divide symmetrically into a pair of CSCs or DCs or asymmetrically into one CSC and one DC. The average number of CSCs that are produced in a division is regulated by an activator of concentration a, whereby we adopt the principle that a high value of a leads to increased production of CSCs, while small values of a enhance the symmetric DC channel. This mechanism can be modeled by a smooth probability function p(D,a). In addition, the model we consider includes a plasticity mechanism for DCs, which are able to de-differentiate and return to the stem state when the CSC density is small. This factor implies a reproduction loop mediated by chemical activators within certain ranges and accounts for the phenomenological evidence of tumor relapses, months after successful therapies. Along these lines, we introduce a plasticity parameter m, which represents the number density of a single plasticity mechanism activator. We use thereby the assumption that the different chemicals produced in the various plasticity pathways are not affecting each other significantly and can hence be (linearly) condensed into one single variable. We can thus build a system that includes four dynamical variables, corresponding to the two cell concentrations *S* (for CSC) and *D* (for DC), and two chemical activator densities: a, the density of stem division activator and m, the plasticity activator.

Differently from many ecological models, the cell dynamical variables are the cellular densities, which correspond to the ratio of the number of cancer cells $N_{D}$ or $N_{S}$ divided by the total cell number *N*, leading to $D=N_{D}/\left(N_{S}+N_{D}+N_{in}\right)$ and $S=N_{S}/\left(N_{S}+N_{D}+N_{in}\right)$, where $N_{in}$ represents the number of inert cells. The inert cells include healthy cells, quiescent or dormant cancer stem cells [20,21] which do not affect the present scenario. The quiescent cells may be in a state of mitotic dormancy or have a weak mitotic activity compensated by apoptosis [22,23,24]. However, they can eventually recover activity and start proliferating, leading them to be considered as the major cause of relapse post full treatment [25]. Since the density associated with $N_{in}$, which we call $N_{0}$, is trivially recovered as $N_{0}=1-S-D$, we eliminate this trivial relationship from Equation (Equation 1). Such representation in terms of density is well suited for a developed tumor containing a large number of cells and allows to implement quite easily the spatial cellular distribution in a second step [5,20]. Contrarily to cells, the activators, inhibitors and growth factors have low molecular weights, thus they do not participate in the mass balance. The system of nonlinear coupled differential equations, generalized from [5] (by making the mitotic rate explicit as well as distinguishing between the activator decay rates) is displayed in Equation (Equation 1).
(1)$$\left\{\begin{array}{cc} \frac{dS}{dt} & =(2p(D,a)-1)\Upsilon \lambda S+q(m)D\\ \frac{dD}{dt} & =2[1-p(D,a)]\Upsilon \lambda S-[d+q(m)]D\\ \frac{da}{dt} & =a\left(\beta S\frac{a}{1+a}-\alpha_{a}\right)\\ \frac{dm}{dt} & =\gamma e^{-S/S_{0}}-\alpha_{m}m \end{array}\right.\;\left\{\begin{array}{cc} p(D,a) & =\frac{\eta a}{\left(1+\eta a)(1+\psi D\right)}\\ q(m) & =\frac{q_{0}}{2}\left[1+\tanh\left(\frac{m-m_{0}}{s_{q}}\right)\right]\\ \Upsilon & =\tanh\left(\frac{1-S-D}{s_{l}}\right) \end{array}\right.$$

The function p(D,a) encodes the aforementioned reproduction mechanism of the stem cells and quantifies their generation rate. For p(D,a)∼0 most CSCs divide symmetrically into two DCs, while for p(D,a)∼1, i.e., as the activator a reaches saturation, a CSC will likely divide symmetrically into two CSCs. It can be seen easily how this feature is upheld by the specific form of p(D,a). The pathway induced by *m* is mediated through an approximated step function q(m), positive for $m>m_{0}$, and vanishing otherwise. q(m) is a measure of the de-differentiation rate due to plasticity of a DC transforming into a CSC as soon as the CSC cell population falls below a critical value $S_{0}$. $\alpha_{a}$ and $\alpha_{m}$ represent the usual uptake chemical coefficients, which quantify the rate of spontaneous decay of the different activators.

Contrary to a previous work [5], we introduce the parameter λ in the two first equations of our system. When λ=1, it means that we choose as a time unit the inverse of the mitotic rate of the CSCs. It also implies that the time scale of our figures has to be compared to this unit of time. However, since this parameter is also a stochastic variable that can vary due to the inherent stochasticity of the cancer cells, this parameter is just a reminder that it must also enter into the stochastic study. Each time we ignore stochasticity, we naturally choose λ=1. The function Υ has been added to the system for the first time in the present work and provides a mechanism to limit the arbitrary proliferation of living cells, that is to encode the domain condition for the cell concentrations of 0≤S+D≤1. In fact, in past works, some trajectories of the variables *S* and *D* could leave the physical domain, given they are relative concentrations, differently from a and m, that are simple number densities. We here assume thereby that the quiescent fraction $N_{0}$ can decrease arbitrarily close to zero. The functional form of Υ through the hyperbolic tangent fulfills this role. Chosen for its simplicity Υ approximates a step function: its value is very close to 1 as S+D is not close to 1, with a negligible effect on the dynamics, whereas if S+D≈1 its value will drop to zero. It should thereby be associated with the mitotic coefficient λ, since the real mitotic rate is the product of λ, the average rate for small densities, with Υ, that accounts for the limit of large densities. The dynamics of each family of cells *S* and *D* is similar, and based on three separate components: a positive production term, linear in *S*, that encodes the mitosis of CSCs; a negative death term, *d* for DCs, which is assumed negligible for CSCs; and a plasticity term q(m)D, encoding the de-differentiation mechanism of DCs.

The activators a and m, on a similar pattern, undergo each a specific production mechanism, and follow an exponential decay rate. In particular, the a production channel is linear in *S*, which provides a dangerous feedback loop. A partial saturation mechanism for large a is expressed with the factor a/(1+a). The production of m is once again strongly coupled with *S*; however, this time the production is enhanced for small *S*, according to the underlying plasticity mechanism, which is triggered for very low concentrations of *S* of order $S_{0}$, the latter being a parameter which will typically be set quite small, about 0.038. Realistic values for the various parameters are set with help of literature data [26], see Table 1. The equations for the different cell families are constructed in such a way that ensures the total generation rate $\dot{S}+\dot{D}$ is equal to the number of divisions of *S* in a unit of time (meaning one stem cell divides into two cells, adding one cell to the balance of the system). Once we assume the stem cells are immortal, the only channel to reduce the total cell population is through the DC death/apoptosis, occurring with rate *d*, a strategy which may be used in therapeutics, see Section 4.

### Static and Dynamics of the CSC Nonlinear Dynamical System

This work draws from past research in the field of CSC models [5,6]. Some interesting mechanisms of stem cell systems have been explored, which also find confirmation within the framework of this research, such as the existence of quiescent states of tumors, the over-shoot of cancerous population post-initiation and before extinction, or the so-called tumor growth paradox, for which larger DC death rates may lead to enhanced proliferation of the tumor, because of competition between the two sub-populations [36]. A dynamical and spatial analysis of the system in Equation (Equation 1) are present in the literature [5]. The present analysis is limited to the dynamical aspects but can be incorporated into a spatio-temporal framework following the steps described in detail in [20].

Work by Olmeda et al. provides a study of the fixed points and their stability and considers two different sets of parameters to discuss the consequent dynamics. A fixed point of a dynamical system is a very specific solution of this system that we call $F_{k}=\left\{S_{k},D_{k},a_{k},m_{k}\right\}$ where the time derivatives are equal to zero, which implies that there is no time variation, meaning the system Equation (Equation 1) remains static in that fixed state indefinitely. A fixed point can be stable (or attractive), meaning that the system labeled by the time-dependent state F(t) will fall onto it if it is close enough: it is also called an attractor; otherwise it can be unstable (or repulsive), meaning any neighboring trajectory will be pushed away from the fixed point. In this case, the trajectory (that is the time-variation of the dynamical variables) depends on the initial data and on the parameters of the dynamics, see Appendix in [5].

The system in Equation (Equation 1) has two or three fixed points, $F_{i}$, i=1,2,3, arising and behaving differently in different regimes of the parameters. From the third equation of Equation (Equation 1), when a(t)→0 it follows p(D,a)→0. A simple computation shows that there exist two fixed points corresponding to this case, $F_{1}$ and $F_{2}$. The first is simply describing the complete elimination of the tumor, with values $F_{1}=\left\{0,0,0,\gamma /\alpha_{m}\right\}$. The second fixed point corresponds to a stable, self-sustaining colony where both $S_{2}$ and $D_{2}$ have constant and nonvanishing values, but $a_{2}=p=0$, giving $F_{2}=\left\{S_{2},D_{2},0,m_{2}\right\}$. Combining the first and second equations of the system in Equation (Equation 1) leads to $D_{2}=\lambda S_{2}/d$ and $m_{2}=m_{0}+s_{q}$arctanh$\left(2d/q_{0}-1\right)$, under the existence condition $0<d<q_{0}$. Then, from the $m_{2}$ value we can compute $S_{2}$ from the last equation of Equation (Equation 1) and get the following: $S_{2}=-S_{0}\log\left(\alpha_{m}m_{2}/\gamma \right)$, with $m_{2}<\gamma \alpha_{m}$, for the concentrations to be real and positive. Therefore the fixed point $F_{2}$ exists if and only if these two conditions are satisfied, while $F_{1}$ always exists. In addition to these two fixed points, there is a case where a third fixed point $F_{3}=\left\{S_{3},D_{3},a_{3},m_{3}\right\}$ arises, the only for which a≠0, and which has no explicit analytic form, since it is obtained by solving approximately a third-degree polynomial.

When approaching a fixed point $F_{k}$, via numerical means, the corresponding set of variables F(t)={S(t),D(t),a(t),m(t)} will evolve in time as $F\left(t\right)∼F_{k}+f_{k}e^{\delta_{k}t}$, where the tiny quantity $f_{k}$ depends only on the parameters. Through evaluation of the Jacobian of the right-hand side field in Equation (Equation 1) at the fixed points, and by looking at the signs of the real parts of its eigenvalues $\delta_{k}$, we can determine whether the fixed points are stable or not. If all eigenvalues are negative the fixed point is stable, while if there is at least one positive eigenvalue it is unstable. For the purpose of analyzing the nature of the fixed points, let us assume for simplicity that the condition $m_{2}<\gamma \alpha_{m}$ is satisfied. We then look at four cases as we vary the parameters *d* and $q_{0}$, and let the others be fixed: namely for $d≶q_{0}$ and d≶ψα/β. The first condition decides whether $F_{2}$ exists or not; in case it does exist it is attractive and stable, and $F_{1}$ is unstable (see Figure 1b), whereas in case $F_{2}$ does not exist, $F_{1}$ is attractive and stable (see Figure 1a). From the viewpoint of the clinics, $F_{1}$ describes a reasonably optimistic situation if no environmental perturbation occurs as time goes on. For certain combinations of parameters, $F_{2}$ may indicate a stable, dormant tumor with low concentrations, upheld by a mitosis-plasticity feedback loop. However, as we will see, in many situations the concentrations of cancerous cells can be stable and relatively large and thus worrisome from a clinical perspective. The second condition rules whether $F_{3}$ exists, which is the case for $d>\psi \alpha_{a}/\beta$. If $F_{3}$ exists, it is always a saddle point and it is unstable.

To summarize, there are either two or three fixed points depending on the parameter values, one of which is always stable, which means trajectories starting sufficiently close to it will converge more of less quickly depending on the existence of an overshoot: compare Figure 1a,b. The fact that $F_{3}$ is never stable makes it less interesting from a dynamical perspective since it is unlikely that a tumor will evolve close to that state. It should be noted that the Jacobian eigenvalues have been determined assuming Υ≈1, i.e., well below the saturation regime. This is a reasonable assumption since we typically use a relatively small value of $s_{l}$ in Equation (Equation 1), and hence expect the influence of the limiting function to be significant only quite close to saturation. Would we consider the full expression of Υ to determine the fixed points, we would have to give up the explicit form for the fixed points for $F_{1}$ and $F_{2}$ as well.

In the following, we give the numerical values of parameters for the results and figures presented here. Changing parameters in the system Equation (Equation 1) allows us to explore the dynamics of the model but some remains fixed in our study such as λ=1.0 except when stochasticity is introduced, ψ=1.0, $S_{0}=0.038$ and $m_{0}=0.5$. The coefficients $s_{q}$ and $s_{l}$ must remain small but their precise values are less important and, respectively, $s_{q}=0.01$, $s_{l}=0.1$ in the following. Variable constants in our study are indicated in Table 2 and Table 3.

## 3. Stochasticity

All the components of a multi-cellular colony present stochasticity, which can be intrinsic or extrinsic depending on whether it is induced by genetic or epi-genetic variations or is due to the micro-environment. Clearly, a solid tumor that results from genetic mutations of the cells is by nature a time-dependent stochastic assembly, and the approach based on cellular density aims simply at establishing an average behavior of the system as time goes on. Nevertheless, the role of stochasticity is not always fully intuitive in ecology dynamics. For instance, it may induce the full elimination of a community by suppressing a sub-population that has resisted exterior stress, as in the case of the so-called Anthropocene extinction [37]. Alternatively, stochastic factors may contribute to the extension of the lifespan of a community [13]. In these two cases, for reasonable values of the noise amplitude, stochasticity works in conjunction with stress applied to the whole community and with a fluctuating perturbation. Our aim is to investigate the relevant processes or dynamical variables that are most sensitive to the noise inherent to any biological system.

In many nonlinear dynamical problems of biology, one deals with macroscopic observables which are often resulting from averaging over very large pools of microscopic variables, the mostly intuitive justification thereof being that these effects due to local variabilities are compensated in average. However, for many systems, the impact of environmental or internal variabilities on the dynamics is often amplified due to the non-linearity of these systems, which consequently requires accounting for these variabilities from the beginning [38]. The direct way to do so is to consider a parameter describing a given mechanism that is affected by the environment as a stochastic variable itself, passing from the “time-averaged” and constant quantity to a stochastic and time-dependent variable of the system. Noise terms have been included in several otherwise deterministic dynamical systems, and it has been shown how this impacts the dynamics to an important extent, for instance by stabilizing regions of phase space that produce diverging trajectories in the deterministic case [8,11]. The extension of deterministic population models by means of stochastic contributions is hence reasonable to the extent that it makes the model more realistic and predictive given the typically large amount of environmental factors that impact individuals in the population. For the case of cell populations such factors have been extensively modeled and embody several phenomena typical of biochemical processes in cells. These include among others diffusion models for chemicals in the cells, stochastic gene expression in cell division and active cellular transport [39]. Furthermore, work by D’Onofrio et al. investigated the impact of stochastic noise in some cell population models that also apply to tumors [40,41].

The mathematical background that is needed to implement stochastic quantities in our dynamical system is part of the rich and vast theory of stochastic, and particularly of Markovian, processes, defined as a family of random variables indexed by a time parameter, such that the probability of an event at a given time is independent of the states at precedent times. In other words, a Markovian process has no memory of past states. Non-Markovian processes, which in turn exhibit lagged response to variations, have been employed to describe different biological and biochemical systems [42]. At the same time, non-Markovian dynamical systems can easily produce bifurcations or attractors, also exhibit chaotic behavior, and are therefore quite complex from a mathematical standpoint. In the present work, we will impose the strong assumption of the absence of memory (i.e., the Markov property) in the system. This simplification is motivated by the significant step-up in the mathematical complexity of more general non-Markovian models, which nonetheless may be an interesting direction for future works.

Stochastic processes coupled to an otherwise deterministic dynamical system can be described by a system of stochastic differential Equations (SDEs). The general form of an SDE describing the time evolution of a quantity *X* is the following: $dX_{t}=\mu \left(X_{t},t\right)dt+\varsigma \left(X_{t},t\right)dW_{t}$. The equation is in differential form as is common for SDEs; in particular, the term $dW_{t}$ represents the variation along a random walk $W_{t}$, parameterized with the usual time *t* and evolving according to a given rule. By convention we call *drift term* the function $\mu (X_{t},t)$ and *diffusion term* the function $\varsigma (X_{t},t)$. In the present work, we pick the Wiener process to let the stochastic parameters evolve: it is the mathematical realization of a Brownian motion, which has typically no memory. It is found in a number of different physical mechanisms, such as the path of a particle in the air, subject to frequent collisions with other particles that tend to randomize the motion. A Wiener process is defined in a mathematical framework as a walk whose increments (i.e., displacement in each time step) are random variables that are normally distributed. A disturbance that follows a Wiener process is said to produce a white noise signal, which means it has constant power spectral density, i.e., there are no preferred frequencies of oscillations. Once again, this choice shall be considered as the precursor of more complex noise models, which require a more advanced treatment but could better fit the actual biology, as argued in [40].

### 3.1. The System with Noise

In order for our dynamical model to enclose some underlying biological processes, we will include stochasticity in the system in Equation (Equation 1). This stochasticity is manifest in various mechanisms that depend on a rapidly varying environment and shall be applied to the parameters that are meant to embody said mechanisms. To build the stochastic differential equation with a parameter subject to noise we take the deterministic system and extend it by means of the stochastic equation which lets the said parameter evolve according to the rule of a Wiener process of given amplitude. For instance, as we apply a simple noise on the parameter λ, the system in Equation (Equation 1) will have the following additional (stochastic) differential equation to describe the evolution of λ: $d\lambda_{t}=\rho dW_{t}$. The diffusion term is hence constant, parameterized through a constant ρ, and there is no time dependence. This extra stochastic equation has such a simple structure to avoid making a further assumption about the model of the noise term. λ has a value equal to $\lambda_{0}$ at t=0 and for each simulation will evolve in time according to a random instance of the Wiener process, with fixed amplitude ρ, which quantifies the expected deviation from the initial value $\varepsilon \left(t\right):=|\lambda_{0}-\lambda \left(t\right)|$. Larger ρ means ε(t) will on average grow faster with time. To make our system of equations consistent after the introduction of the noise on λ, we must look wherever λ appears as a constant in Equation (Equation 1) and replace it with the time-dependent λ(t). This entire paragraph is valid for any other parameter different from λ on which we wish to apply the noise to investigate its impact. To obtain a significant result that does not depend on the specific instance of the stochastic process we shall simulate the system repeatedly, say *N* times, and then take the average of the obtained trajectories. This method to study the system with stochasticity requires making some choices. The first consists of the selection of the parameter which shall undergo stochastic variation, probably the most important in the system biology; secondly, we need to fix a value of the noise amplitude; finally the number *N* of trajectories we consider and average.

As we see in Equation (Equation 1), and despite the biology of CSCs being oversimplified, our system is described by a quite large number of parameters. However, it is clear that not all of them have a particular physical significance: some of them, such as $s_{q},s_{l}$, are rather mathematical artifacts needed to provide an optimal description of a mechanism. Our analysis suggests that it is reasonable to distinguish between three subsets of parameters, ordered by their biological and physical relevance as well as their impact on the system. The first consists of those parameters that have a clear physical and biological meaning and interest, and which can be reasonably impaired by environmental fluctuations, namely the mitotic rate λ, the DC death rate *d* and the plasticity coefficient $q_{0}$. These three concern directly the cells. There are many examples of mechanisms that are impacted by external fluctuations, ranging from physical variables, such as local temperature variations and pressure gradients affecting cell transport, the micro-environment of the assembly, to biological mechanisms such as genetic variability, affecting among others mitotic and apoptotic rates. The second class of parameters concerns the mechanisms related to the chemical activators or inhibitors, hence having yet a significant bio-physical meaning, but a smaller relevance in the framework of this research. In addition, they have lower chances to be investigated experimentally, which hinders the possibility to test our assumption about the way the environment affects the said parameters. They are η and ψ, and the activator decay rates $\alpha_{a}$ and $\alpha_{m}$. The rest of the parameters are those with little experimental feedback and are generally less likely to be affected by environmental fluctuations.

The noise amplitude that we use for investigating a single parameter must be chosen carefully and in line with the average value of the parameter, under the assumption that the fluctuations due to a varying environment are small compared to the parameter magnitude. Generally, we will work with amplitudes between 5% and 20% of the parameter values. This model has some limitations, which become more visible if we increase the relative amplitude up to a point where the value of the parameter will likely become nonphysical within our typical time scales (t∼10–20 d), for instance leading the death rate *d* to take negative values.

To determine the optimal number of trajectories, we need to take into account several factors, that include the accuracy of the mean as well as the computation time. On the one hand, generating more trajectories enables us to reduce the impact of the single trajectory deviating too much from the average, and ensures that if we repeat the experiment we obtain the same result. At the same time, a large *N* value increases the likelihood that there will be extreme trajectories in finite time, such as nonphysical or strongly diverging ones, which may ruin the average completely, and would require a specific treatment [40]. Finally, one shall consider the simulation time, which increases close-to-linearly with *N*. A reasonable pick for *N* is consequently based on the available computational power and on the instance of the system one simulates, meaning the choice of parameters and initial conditions, which affects the likelihood to produce problematic trajectories. Furthermore, considering the scope and aims of this work, which did not require too large statistics, we generally worked with *N* between 20 and 100. Examples of stochastic trajectories are reproduced in Figure 2. Details regarding the stochastic integration are provided in Section 5.

### 3.2. Fixed Points

We now want to evaluate the response of the system to the introduction of stochastically varying parameters. It is reasonable to look at the effect of stochasticity on a situation that remains static in the deterministic case meaning for trajectories close to an attractive fixed point. We can simulate many instances of the dynamical system with stochastically varying parameters and look at the distribution of the trajectories. Performing this process with noise applied to different parameters will enable us to observe quite directly to what extent applying noise to each parameter will affect the dynamical variables in our system. In addition, the method gives the range of noise amplitude which keeps the system reasonably stable on average without producing nonphysical trajectories or large deviations from expected convergence behavior. Figure 3 depicts the time evolution of the distribution amplitude σ(t) of a bundle of stochastic trajectories, in different time ranges.

Here we focused on a typical set of parameters leading to the stable fixed point $F_{2}$. We did not represent the amplitudes corresponding to the variable a, since its value is numerically zero, which implies that there will be no impact due to stochastic parameters. From Figure 3b we see that, within a time range comparable to the characteristic relaxation time of the system, i.e., t≲10d, the deviation σ(t) of the stochastic bundle of trajectories around $F_{2}$, for different stochastically varying parameters and dynamical variables is linear with a good approximation. For a fixed choice of variable and stochastic parameter, steeper lines mean the said variable is strongly affected by the noise on the said parameter. We notice that the variable *D* is overall the more strongly impacted by stochasticity, particularly for the parameters λ and *d*, while the slope for *S* is a lot smaller within the range t∼10d. The fact that σ(t) grows linearly, or even just according to a power-law, is a good indication of the stability of the model in presence of stochastic events since such growth is subdominant compared to the exponential growth which follows from unstable trajectories. As a term of comparison, the deviation for a simple stochastic Wiener process follows a square-root rule. How this rule of the noise term affects and translates in the variables of the dynamical system is a consequence of the form of its equation and particularly of nonlinearities in the model. As one could expect, when considering longer time ranges, the stochastically varying parameter may possibly take non-physical values. This has problematic consequences on the dynamical variables, such as leading to the concentrations leaving the boundaries of physically meaningful processes. There are a variety of possible solutions to this issue, which may include a pre-selection of the stochastic trajectories, as well as extending the model for the stochastically varying parameters, for instance by the addition of dynamical terms that avoid large deviations from the base values. For the purpose of this first analysis, we chose to keep our simplest model of a Wiener process, since the parameters and the subsequent time scales we have considered were within the range of stability of our methods. Tools to extend the validity of the stochastic model onto longer time intervals could be the subject of future works.

### 3.3. Discussion

We investigated the system given in Equation (Equation 1) after applying a white noise term to some key parameters of the model, in order to account for variabilities due to the environment and other factors [43]. Three main parameters have been looked at closely, in order to understand and quantify the impact of the noise on every single dynamical variable, namely the stem cell reproduction rate λ, the differentiated cell death rate *d* and the amplitude in the plasticity mechanism $q_{0}$. We allowed the said parameters to evolve stochastically according to the rule of a Wiener process and considered the consequent response of the system. To quantify the response for a specific parameter we considered the distribution of a bundle of stochastic trajectories over time, and the width thereof, obtained through an un-binned Gaussian fit. We observe this width grow linearly, steeper for some parameters and variables, up to an approximate stability threshold, where we observe sharp, nonlinear increases. According to our analysis, this threshold is, up to some extreme cases away from typical parametric regimes, located above the characteristic response time scale of the system (estimated by means of the relaxation time onto stable states). The latter depends on the choice of the base parameters, a few of which constitute a rate and therewith fix a time scale. We can conclude that the system is essentially stable after introducing stochasticity, with the reasonable condition that we remain within a time range comparable to the characteristic time scale of the system.

There is another criticality that should be considered with respect to the introduction of stochastic parameters. Namely, as we analyze the system by looking at the mean trajectory from a bundle thereof, we are assuming that the variations of the stochastic trajectories with respect to the deterministic case are of order with the amplitude of the noise on the stochastic parameter. In other words, we presume that, although each trajectory undergoes certain small fluctuations, the general qualitative behavior will be the same as in the unperturbed case, and thereby fluctuations tend to compensate as we take the mean. It may be the case, however, that our initial conditions, normally leading to a stable state, are close enough to a point where the system is unstable and leads to uncontrolled proliferation (one can usually augment a(0) up to a point where this is the case). Therefore, even if most stochastic trajectories are going to converge on, say, $F_{1}$ or $F_{2}$, there may be some rare divergent cases that follow a fundamentally different path and strongly affect the mean value. Clearly taking the mean will be problematic in this case, and one will observe it quite differently from the unperturbed case. One should therefore be aware of this possibility, which does not quite represent a weakness or a vulnerability of the system before stochasticity, as much as a natural consequence of the fact that there exist unstable trajectories, and that they can take place due to stochasticity. Studying the system in the bordering region between per se stable and unstable trajectories entails the possibility of passing from one regime to the other. These unstable trajectories shall be rather seen as a (small) class of cases where this stochastic analysis necessarily loses its significance and descriptive power.

## 4. Treatments

One essential topic within the context of cancer cell population models is the investigation of drug treatments. From a pharmacological perspective, the treatment of a patient is the fruit of a complex trade-off between a number of factors, such as dose, exposure and the effects on the body positive or negative that are specific to the employed drug. The job of determining the most effective therapy for single patients is made even harder by the intrinsic variability between patients and typically nonlinear drug kinetics and dynamics. The need for a clear understanding of these intertwined mechanisms relating pharmacology to living organisms led to the emergence of the so-called quantitative systems pharmacology (QSP) [44]. This research area combines the deterministic and average-based mathematical-biological models with pharmacometrics involving also models for system variability and accounting for the kinetic and dynamic mechanisms of drug action [17].

When modeling the interaction with the host organism of an administered substance within a therapy, one has to distinguish between two distinct mechanisms, generally indicated as the kinetics as opposed to the dynamics of the drug. With drug kinetics, or pharmacokinetics (PK) one wants to address the principles that regulate the time evolution of the drug concentration in the blood. On the other hand, with drug dynamics, or pharmacodynamics (PD), we mean quite directly the effect that the drug has on the body and on the tumor. That corresponds in practice to the rule to implement the drug effect in our system’s dynamics with respect to the time-dependent drug concentration in the organism. Extensive discussions of the main used PK and PD models can be found in the literature [18].

There exists a distinction between the two main classes of cancer treatments, namely that between cytotoxic and cytostatic drugs. The former is what is usually meant by the term chemotherapy and consists of substances aimed at killing the cancer cells directly. On the other hand, with cytostatic drugs, we understand a class of drugs that inhibit cell division and in general slow down cell proliferation, for instance by targeting activators and growth factor receptors [16]. Furthermore, we will consider other less common but promising types of treatments, which involve interfering with the plasticity and the differentiation mechanism of CSCs.

### 4.1. Pharmacokinetic Model

We build our pharmacokinetic (PK) model starting from the basic assumption of the regular and periodic administration of a specific drug dose. We use a one-compartment model, which means we assume a homogeneous concentration and elimination rate of the substance in the organism. The level of drug in the organism χ(t) results from a periodic dose administration which lasts for a time τ and leads to the following differential equation:(2)$$\left\{\begin{array}{c} \dot{\chi }\left(t\right)=\zeta -\alpha_{d}\chi \left(t\right)\;\mathrm{if}\;nT\leq t<nT+\tau \\ \dot{\chi }\left(t\right)=-\alpha_{d}\chi \left(t\right)\;\mathrm{if}\;nT+\tau \leq t<\left(n+1\right)T \end{array}\right.$$
where ζ represents the drug amount injected divided by the administration time τ.

In Equation (Equation 2) *n* labels the *n*th administration phase. Equation (Equation 2) can be solved analytically and describes reasonably the actual mechanism of diffusion of the drug in the body, its absorption by the organism and hence its decay with a characteristic timescale $1/\alpha_{d}$. The period *T* or the rest period T−τ are significantly larger than the drug decay time, due to a high clearance rate $\alpha_{d}$. This avoids drug accumulation within the organism between successive daily administration sessions. For simplicity, we assume the amount of drug in the organism χ to be a dimensionless quantity, while the standard unit of time is one day. In all figures displayed here, the time scale is the day. The value for $1/\alpha_{d}$ is estimated in the order of minutes to hours [18]. It may be useful to express the relationship between the administered dose per unit of time ζ and the subsequent maximum concentration of therapeutic agent being in the organism at time τ that is $\chi_{max}=\chi \left(\tau \right)$. Solving Equation (Equation 2) for the reference interval t∈[0,τ] gives $\chi \left(t\right)=\zeta \left(1-e^{-\alpha_{d}t}\right)/\alpha_{d}$. The subsequent value for the drug administration rate ζ as a function of the desired peak level of the drug is then $\zeta \left(\chi_{max}\right)=\chi_{max}\alpha_{d}/\left(1-e^{-\alpha_{d}\tau }\right)$. We have thus far constructed a model that describes the time evolution of the active drug chemical concentration within the body, which means we have the tools to explain how the body processes and eliminates the substance. To evaluate how the substance affects the tumor within our dynamical system we now want to formulate a model of the drug action.

### 4.2. Pharmacodynamics

This section is devoted to the drug’s effect on the body and specifically on the cancer cell colony. The drug efficacy does not depend solely on the value of χ, since saturation effects may occur for large amounts of the drug, while a very low level may be irrelevant for the cell colony. For intermediate values of χ, the function of drug action Ω(χ), encoding its action onto the dynamical system of the cancer evolution, could in first approximation be linear, as $\Omega \left(\chi \right)=\Omega_{0}\chi$. A more realistic model for Ω(χ) is an approximate step function that produces plateaus for very high and very low concentrations χ and is linear for intermediate values. This approach is displayed in Equation (Equation 3).
(3)$$\Omega \left(\chi \right)=\frac{\Omega_{0}}{2}\left(\tanh\left(\frac{\chi -\chi_{0}}{s_{\chi }}\right)+1\right)$$

As we adopt such representation for the drug action, we are adding three new parameters to the model, $\Omega_{0}$, $\chi_{0}$ and $s_{\chi }$. They should be chosen in a way to address the considerations above about the action for very small and very large concentrations. The parameter $\chi_{0}$ provides a reference for the center of the linear regime, and should be calibrated in dependence of the dose or equivalently of $\chi_{max}$. The dimensionless parameter $s_{\chi }$ can be tuned to determine the width of the linear regime and the points at which the effect of the drug saturates for large concentrations. The fact that the drug concentration produces an effect only through the action function suggests a degree of arbitrariness in the choice of the dose ζ, since we define a range of variation of the action in the variable χ through the parameters $\chi_{0}$ and $\Omega_{0}$, while the amplitude of the action is defined only by the parameter $\Omega_{0}$. In other words, once we fix $\chi_{0}$ and $\Omega_{0}$, we have set the minimum and maximum values of χ, below and above which a variation in χ does not result in a variation in the action, corresponding to the very small and very large concentration cases we discussed above. The value of $\Omega_{0}$ enables us in practice to decide upon the impact of the drug on the parameters we are considering. This small redundancy is however justified by the simplicity of the resulting model in its practical uses, also in the perspective of measuring the free parameters through clinical experiments. A realistic choice of parameters, which is the one we will use throughout most of this study, is provided in Table 4 and produces a behavior that is given in Figure 4.

#### 4.2.1. Cytotoxic Therapy

With cytotoxic therapy, we understand a treatment that targets and kills the cancerous cells directly. Since in our model we have assumed the CSCs as immortal, such a treatment aims at killing the DCs, which corresponds in dynamical terms to affecting the DC death rate parameter *d*. The system we thus solve in this framework is a version of the system in Equation (Equation 1), in which the parameter *d* becomes time-dependent and equal to the sum of the base value $\tilde{d}$ and the action factor Ω(t) from Equation (Equation 3), i.e., in Equation (Equation 1) we substitute the constant *d* with $d\left(t\right)=\tilde{d}+\Omega \left(t\right)$.

In Figure 5a–d, we find examples of trajectories after cytotoxic drug therapy. In Figure 5a–c, we observe an optimistic case where the cytotoxic drug brings the tumor to regression at different rates due to different intensities of the treatment. Parameters of the modeling for Figure 5 can be found in Table 2, Table 3 and Table 4. The untreated case corresponds to trajectories converging rapidly towards the fixed point $F_{2}$, that is the state of stable, self-sustained feedback loop between stem and differentiated cells, due to plasticity. The treatment action, by increasing periodically the parameter *d*, leads the system to break this worrying feedback loop, and effectively converge towards $F_{1}$, meaning tumor extinction. The time scale of such regression clearly depends strongly on the dose ζ and on the action intensity $\Omega_{0}$. A different situation is displayed in Figure 5d, where we observe the so-called tumor growth paradox. The untreated case corresponds to a dangerous overshoot, due to a large initial value of stem activator a, which however falls autonomously after a certain time. The cytotoxic treatment, on the other hand, leads to a fall in the DCs due to an enhanced death rate, which consequently allows a faster proliferation of CSCs because of competition between the populations. This small enhancement of *S* is enough to activate the feedback between CSCs and the activator a, which rapidly leads to uncontrolled proliferation of the tumor.

#### 4.2.2. Cytostatic Therapy

With cytostatic therapy, we understand a class of cancer treatments that aim at slowing down tumor proliferation by targeting the underlying activators and division mechanisms, instead of killing the cells directly, as in the case of cytotoxic drugs. Cytostatic drugs present considerable druggability and good pharmacokinetic properties, involving small molecules. Furthermore, when they concern CSC pathway inhibitors, they do not compromise the healthy cells of the host organism. Since many pathways are critical for CSC progression and self-renewal, in addition to resistance to treatment, the search for multiple-signaling pathway inhibitors has led to the execution of various pharmacologic and clinical studies recently, while further investigations are crucial if one considers the diversity of cancers and of target organisms. Although many cytostatic treatments are on the market [32,45], their efficiency depends significantly on the time of administration, either in early clinical trials or as a maintenance therapy to avoid relapse and metastasis. As for what concerns our model, the reproduction mechanism is entirely due to stem cell mitosis, hence a cytostatic therapy would diminish the mitotic rate coefficient λ. Similarly, as for the cytotoxic therapy, this results in subtracting the time-dependent action Ω(t) from the base parameter $\tilde{\lambda }$, which implies we replace the instance of λ in Equation (Equation 1) with $\lambda \left(t\right)=\tilde{\lambda }-\Omega \left(t\right)$. The parameter λ is quite relevant from a dynamical perspective since it provides an essential time scale, which constitutes a reference for the other parameters. The rationale of this type of therapy lies thus in the dilation of the characteristic growth time scale of the tumor, to consequently enable existing regression mechanisms. Two examples of trajectories undergoing cytostatic therapies are displayed in Figure 6.

Figure 6a shows the trajectories resulting from a successful cytostatic therapy, compared to the untreated case, which leads to uncontrolled proliferation and completes the overtake of CSCs. Such parameters lead to the existence of the only stable fixed point $F_{1}$. The consequence of the therapy in this case is to conduct the otherwise unstable system into a regime where the tumor undergoes regression. In Figure 6b, we have an example of unsuccessful, and even deleterious, cytostatic therapy. The untreated trajectories do in fact describe an autonomous regression of the tumor, which could for instance be caused by an immune system reaction, while the treatment determines a delay in the regression, due to the dilation of the system’s characteristic time scale produced by the reduction of the mitotic rate.

#### 4.2.3. On Plasticity

Another type of treatment we investigate is relevant for the current research since it targets the plasticity of tumor cells [46], which is the feature the present model has been extended by, starting from the existing CSC models [5]. It probably constitutes the most innovative possibility of treatment among those we investigated upon the present model, given the growing interest in the mechanism of plasticity within cancer biology and pharmacology. It may also concern tumor drug-resistant cells which appear after cytotoxic treatments and which exhibit stemness features [47]. Similarly as in the cytotoxic therapy that we considered, the rationale of treatments targeting plasticity is to break the feedback loop between stem cell differentiation through mitosis, and de-differentiation of DCs back into stem state. This feedback is in fact the mechanism responsible for the self-sustained state which we identified with the attractive fixed point $F_{2}$. In this type of therapy, we subtract the drug action Ω(t) from the parameter $q_{0}$ with base value $\tilde{q_{0}}$. In formulae, we trasform $q_{0}$ in $q_{0}\left(t\right)=\tilde{q_{0}}-\Omega \left(t\right)$, subsequently reducing in magnitude the plasticity coefficient q(m). The latter is activated quite steeply through an approximated step-function as the activator density m approaches the value of $m_{0}$. As a consequence, an action on the parameter $q_{0}$ has an impact on the system only as long as m stays large enough to activate the plasticity mechanism. There exist thereby a number of possible outcomes of such a treatment, that depends both on the initial conditions, on the system parameters and on the drug parameters.

An optimistic case is depicted in Figure 7a. The corresponding deterministic system is attracted by self-sustaining state of $F_{2}$, which can be potentially dangerous, according to the fixed point existence conditions. Since $q_{0}$ and *d* are relatively close to one another, reducing the value of $q_{0}$ would break the self-sustaining loop and lead to tumor regression, ending in the stationary state $F_{1}$. In the case of the trajectory with drugs, while the value of $q_{0}$ oscillates in time, the drug action is sufficient to determine a net effect of breaking the plasticity loop, hence leading the system to a slow regression. On the other hand, Figure 7b represents a case where the drug action is not strong enough to break the plasticity loop, and besides the short oscillations due to the therapy, the system relaxes stably on the self-sustained state of $F_{2}$. Following the same principles, drugs acting on plasticity are ineffective in case the tumor is undergoing spontaneous regression, possibly after an overshoot. During the overshoot, which is the dangerous part from a clinical perspective, the activator m is below the plasticity threshold, whence the drug is ineffective, and by the time the plasticity mechanism is activated, and with it the drug action, the tumor has already undergone significant regression.

#### 4.2.4. Alternative Therapies

A long-standing suggestion [48] to control the harmful role of CSC is to push them into a differentiated state and simultaneously attack these newly differentiated cells via cytotoxic drugs. This idea, suggested two decades ago, was sustained by the use of RAs (retinoic acids, derived from vitamin A) which lead to gene transcription and differentiation [49], and while it has not given satisfactory results in clinic yet, it has not been abandoned. Indeed, several pharmacologic forms of this method exist and can be of use for specific cancers. Additionally, one may focus on the signaling pathways that were mentioned above, which can be blocked, then also leading to differentiation. Perhaps more interestingly the use of ultrasounds in combination with micro-bubbles containing cytotoxic therapy has shown a noticeable effect on the apoptosis of human breast CSCs in vitro and in vivo [50]. However, it was also observed that liver cancer stem cells were forced to undergo differentiation when subject to dual-frequency ultrasounds. So obviously, ultrasounds will be a precious instrument in cancer therapy. Concerning the mathematical analysis of a treatment forcing the CSC differentiation, it is sufficient to modify the plasticity term q(m) in Equation (Equation 1) by $q\left(m\right)-\Omega_{diff}\left(\chi \right)$, and then the coefficient of death *d* of the DCs by $d\left(t\right)=\tilde{d}+\Omega_{cyto}$ where $\Omega_{diff}$ is the drug function for differentiation and $\Omega_{cyto}$ is the traditional chemotherapy function. In a way, this method could resemble a combination of the cytotoxic and the plasticity-targeted therapies that have been discussed above: the effects on the system of the drug action summed on the parameter $q_{0}$ and on the function q(m) are very similar, while the action on *d* is identical to what we considered in the case of cytotoxic therapies. Some results are given in Figure 8. Being a very interesting strategy for active as well as quiescent stem cells, it appears very promising especially if we can avoid affecting the adult “healthy” stem cells with ultrasounds.

### 4.3. Discussion

To investigate the possible therapies upon our model of cancerous cells, we first developed both a pharmacokinetic and a pharmacodynamic model for the drug, to describe how the substance transits through the host organism, and how the tumor colony responds to it. By means of these models, it was possible to focus on three different types of therapies: cytotoxic, cytostatic and one that targets the plasticity mechanism. The cytotoxic therapy aims at killing the cancerous cells directly, and since we assumed stem cells to be immortal, its effect will be on the differentiated cells, namely an enhancement of their death rate. Among the different possible outcomes of such therapy, we discussed one effective case, where it was able to break the feedback loop that upholds a stable and self-sustained colony, hence leading to tumor regression. A further possible outcome corresponds to the rather dangerous tumor growth paradox, where the drug action itself causes an uncontrolled proliferation of CSCs due to competition between populations. A cytotoxic therapy shall therewith be considered dangerous in case the system is in a phase of fast proliferation, due to a high concentration of stem activator a.

In our approach, CSCs are considered absolutely drug resistant. It is true that they exhibit innate resistance such as DNA damage repair and apoptosis evasion mechanisms. However, they also acquire drug resistance through new mutations and evolve during the drug treatment. In addition, targeting CSC is not obvious [51]: their number is small, they are localized in specific niches and one important aspect of pharmacology consists of identifying them using their specific biomarkers. This reinforces the need for therapies different from cytotoxic treatments, in other words, to target the specific pathways that these cells need to reconstitute their niches as their micro-environments [43]. However, the actions of such pathways as Notch, Wnt/β and Hedgedog [4] are strongly coupled and thereby therapeutics are required to act on several of them at the same time.

Cytostatic therapies are aimed at slowing down the stem reproduction rate. We observed a case where the untreated scenario led to fast uncontrolled proliferation and diverging activator a, and the therapy was thereby effective in bringing the tumor in regression, showing the great potential of such therapy within this model. The WNT signaling pathway is an example of a potentially targetable pathway [52], which appears to be responsible for the maintenance and expansion of stem cell colonies (not only CSC). Being also responsible for the anti-tumoral immune response of the CSC cells, our result suggests it may be the main target of cytostatic drugs in the future. However, since it affects all stem cells of the adult body, it may have serious negative side effects on the health of patients. Besides, there are also cases where cytostatic therapies are ineffective, such as for overshoots [53] followed by spontaneous regression in the untreated case. A therapy that targets plasticity is able to undermine the mechanism leading to the stable tumor colony and bring it to regression. Quite predictably, such therapy is only effective if plasticity is in place, which may not always be the case, and it should therefore be employed possibly only once certain of the state of the tumor. This therapy should be employed after a successful cytotoxic or cytostatic therapeutic period leading to a decrease of the stem and DC cell populations, in order to prevent the relapse of the tumor via plasticity. Indeed this therapy will be ineffective as long as the plasticity pathway remains inactive; however, it may determine the full tumor extinction under specific conditions. Finally, we considered the combined action of a cytotoxic drug and a drug that stimulates the differentiation of stem cells, effectively producing an artificial “death rate” for CSCs. Such a method was effective in the two considered situations, yet investigated for the other types of therapies. In one case it was able to break the self-sustained differentiation-plasticity feedback, similar to the simple cytotoxic therapy as well as the one targeting plasticity: it was of course expected since the present method is the combination of cytotoxic and plasticity-targeted. In the other case, the method was effective in a situation where the cytostatic drug was effective in bringing a rapidly growing stem proliferation to regression. These simple examples underscore the potential of such a combined therapy, although one should keep in mind that a therapy enhancing stem cell differentiation may impact other stem cell colonies naturally present in the organism with undesired consequences. The aforementioned possibility of ultrasounds appears to be particularly interesting in this respect, enabling local treatments directly on the cancerous region.

As we observed, there are a few different possible types of the evolution of the tumor colonies [1], and each may have a particularly suited therapy to be treated with. In case of rapid stem proliferation, signaled by high concentrations of a, the cytostatic treatment is the more effective, while a cytotoxic may even deteriorate the situation. To treat a stable state upheld by plasticity, one shall employ either cytotoxic therapy or possibly target the plasticity and differentiation mechanism, in order to undermine the dangerous feedback loop which supports the state. It is an interesting option to adopt the two therapies jointly. Finally, none of the therapies we examined were effective in treating an overshoot [53] due to a sudden spike in the stem activator. To this end, our suggestion is to develop treatments aimed at controlling the dynamics of the activator a, by enhancing its decay or by inhibiting its production channels. One must also consider that the part of this model describing the dynamics of the chemical activators is likely to be just a simplification of more complex underlying mechanisms, which for the moment do not find phenomenological upshots easily.

## 5. Methods

Simulations have been performed with the support of open-source libraries from the Julia project, which provide among the most efficient tools for numerical integration, both of ODEs and especially of SDEs [54,55]. To integrate the dynamical system in Equation (Equation 1) we tested a number of different numerical methods of different orders. After a thorough comparison, we adopted the Tsit5 algorithm, a Tsitouras adaptive improved Runge–Kutta method of order 5/4 [56], to solve the deterministic system, both with and without drugs. It consists of a refinement of the common ode45 integrator, and it delivered the compromise between high computation speed as well as consistent results when compared to higher-order methods. Ordinary Runge–Kutta methods, such as ode45, can only take deterministic problems in input. Dealing with stochastic differential equations requires the employment of suited methods that depend on the structure of the system as well as on the specific form of the stochastic noise. The fact that the noise we have added to the system is independent from the system variables would in principle have enabled us to first generate a random walk for the stochastic parameters and then solve the noised system as if it were deterministic. However, we conveniently treated the stochastic problem as a general system of stochastic differential equations. Typical adaptive integrators such as Tsit5 would otherwise easily become inefficient since the scale of variation due to the noise is significantly smaller than the typical magnitudes of the derivatives in the deterministic system. Additionally, using specific stochastic solvers has relevant advantages in terms of scalability, since it can be easily adapted to more complicated types of noise, which may for instance depend on the system variables. For the stochastic problem we hence used the sriw1 method, an adaptive, stochastic Runge–Kutta integrator with strong order 1.5 and weak order 2.0. A useful statistical tool that we used to calculate the unbinned maximum likelihood fit in order to quantify the deviation due to stochasticity is the MINUIT routine [57] from the ROOT project [58].

## 6. Summary and Conclusions

The present work has dealt with some fundamental questions about the Cancer Stem Model (CSM), which assumes the existence of different families of cells within a tumor colony, namely cancer stem cells (CSCs), differentiated cells (DCs) and inert cells. CSCs exhibit almost-immortal features, are strongly drug-resistant and proliferate very fast through mitosis into other CSCs or into DCs. Their reproduction rate is regulated by a chemical activator, which generates a feedback loop. Differentiated cells are characterized by a negligible reproduction rate in comparison with CSCs and a significant death rate (i.e., transformation into inert cells). Inert cells do not contribute to the dynamics of the tumor. The present model is distinguished for including a plasticity mechanism, of DCs being able to de-differentiate into stem state when the stem cell concentration is very low. This process is mediated by another chemical activator included in the dynamics. The objective of this research was to investigate the system in the presence of stochastically varying parameters and the response to drug treatments.

Firstly, past research in the framework of dynamical system has employed stochasticity to account for variabilities due to the environment or other mechanisms [43]. We extended thus our model for cancer cells and activators by means of a system of stochastic equations describing the evolution of important parameters of the model under a white noise. We then have considered bundles of stochastic trajectories and quantified their deviation from the deterministic case, providing therewith a measure of the impact on the various variables of stochasticity as applied on different parameters. We could observe that the introduction of stochasticity did not change the dynamics significantly, as long as we remained within a reasonable range of noise amplitude and within the characteristic time scale of the system to relax on stable states. The fact that the system is stable under stochasticity constitutes an important validation of the model that could pave the way to further investigations of some of the mechanisms in place. If one is interested in working with larger amplitudes or longer time scales it may be necessary to extend the model for stochasticity in a way to prevent the onset of nonphysical trajectories.

We then developed a model for drugs both regarding its kinetics and dynamics, which thus described the evolution of the substance in the organism as well as its action onto the tumor colony. We considered different drug action possibilities, which included that of a cytotoxic and a cytostatic treatment, as well as a therapy that targets the plasticity mechanism, and one that combines a cytotoxic drug to enhancement of the stem cell differentiation mechanism. Some important cases have been represented and discussed, outlining situations where the therapies are effective as well as where a treatment is problematic and may even be dangerous. These results, however reliant on the assumptions of the model, may have important clinical implications and may lead to the development of new perspectives in pharmacology and cancer biology.

## Figures and Tables

**Figure 1 cancers-15-00677-f001:**
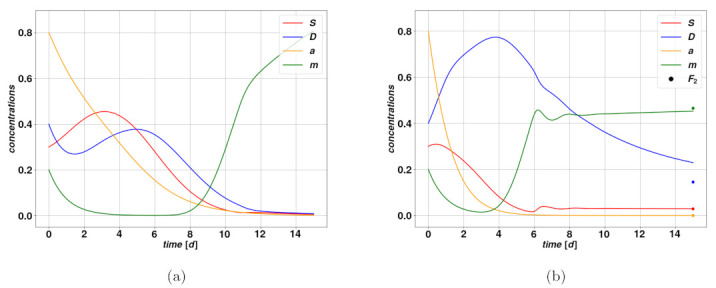
Examples of simple trajectories with an overshoot, converging on $F_{1}$ (left, (**a**)) and $F_{2}$ (right, (**b**)). Lines in red indicate the variable *S*, in blue the variable *D*, in yellow the activator a and in green the activator m. In (**b**), the coordinates of the fixed point $F_{2}$ are given in colored dots. Parameters of the modeling can be found in Table 2 and Table 3. Initial conditions (t=0) differ significantly from the fixed point values recovered at long times t≃10, except for *D*, because of the overshoot.

**Figure 2 cancers-15-00677-f002:**
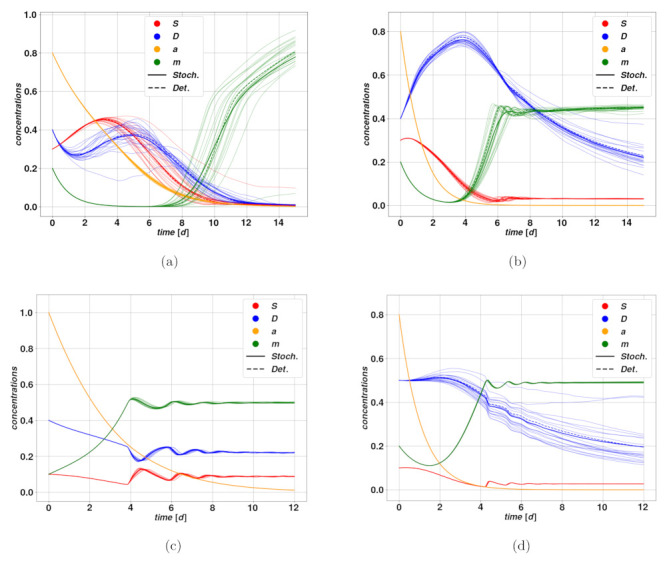
Examples of stochastic and deterministic trajectories. In blue, the concentration *D* of differentiated cells, in red the concentration *S* of cancer stem cells, in orange the activator a and in green m the plasticity signal. All these quantities are solutions of Equation (Equation 1), with the additional stochastic equation describing the stochastically evolving parameter. The stochastic bundle is pictured in thin solid lines, the thick line indicates the average stochastic trajectories and the dashed lines correspond to the trajectories without noise, as a term of comparison. (**a**,**b**) correspond to the same parameters and initial conditions as those in Figure 1, with noise applied on the parameter λ and convergence on the fixed points $F_{1}$ and $F_{2}$, respectively. In (**c**) we have an example of how the plasticity mechanism affects the dynamics: up until ∼4d both *S* and *D* appear to be falling, along with the mitotic activator **a**, which may lead to the complete annihilation of the tumor. However, as the plasticity activator **m** reaches a value about $m_{0}=0.5$, we notice a sudden bump in *S* due to plasticity, followed by damped oscillations around the stable fixed point $F_{2}$. The plot (**d**) has the same parameter values as (**b**), but stochastic noise is applied on *d*, which explains why the trajectories of *D* spread in a wider range compared to the other variables, and to the trajectories in (**b**). In each case the noise comes with a 10% amplitude with respect to the base value, and a bundle of N=20 stochastic trajectories is displayed.

**Figure 3 cancers-15-00677-f003:**
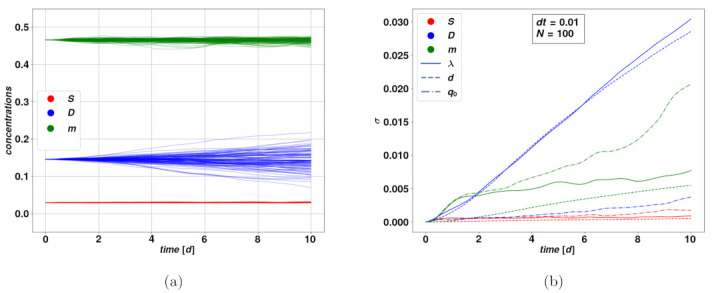
Study of the stochastic trajectories in neighborhood of fixed point $F_{2}$. On the left (**a**) displays a bundle of N=100 stochastic trajectories, whereby the fluctuating parameter is λ. On the right (**b**) shows the accretion of the bundle width over time for the dynamical parameters of the system subject to stochastic noise on each of the parameters λ, *d* and $q_{0}$. The bundle width is quantified by the standard deviation σ of the distribution of the trajectories at each point in time. σ is estimated for each variable using an unbinned maximum likelihood fit of a Gaussian distribution, of which σ equals the fitted standard deviation. As previously, we display in blue the concentration *D* of differentiated cells, in red the concentration S of CSCs, and in green the plasticity signaling concentration. The variable a is not displayed since it corresponds to a numerical zero and undergoes no variation even in the stochastic trajectories. The stochastic bundle is pictured as the thin solid line. The style in **b** depends on the choice of stochastic variable: continuous line for λ, dashed line for the death rate *d* and dot-dashed for the plasticity coefficient $q_{0}$. The fluctuating parameters evolve according to an unbiased Wiener process, starting from fixed base values. In each trajectory, the noise amplitude is of 10% of the base value. We see how most amplitudes grow linearly.

**Figure 4 cancers-15-00677-f004:**
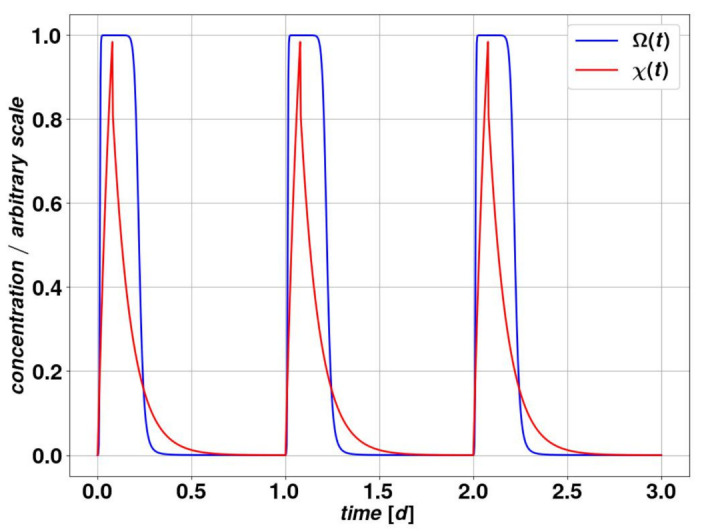
The concentration of the substance in the organism, labeled as χ(t) in blue according Equation (Equation 2) and the action function Ω(t) in red as in Equation (Equation 3) during 3 periods. The parameters correspond to those displayed in Table 4.

**Figure 5 cancers-15-00677-f005:**
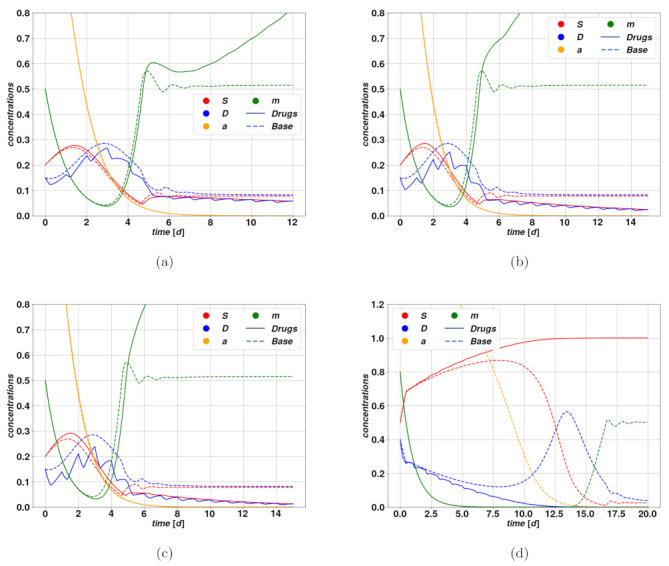
Trajectory overlay with cytotoxic therapy, with drug action on the parameter *d* (solid lines), and case without drugs (dashed lines). In all figures, we maintain the same choice of colors as previously for *S*, *D*, a and m. The system parameters are indicated in Table 2 and Table 3. In three of the figures (**a**–**c**), we display a case where the therapy is effective in breaking the self-sustaining mitosis-plasticity loop, and leads the tumor to regression. The simulations have identical system parameters and initial conditions and only differ in the dose, expressed through $\chi_{max}$ and the action magnitude $\Omega_{0}$ of the drug. In (**a**) we have the usual scheme in Table 4; in (**b**) we changed and used $\chi_{max}=2.0$ and $\Omega_{0}=1.5$; in (**c**) we adopted $\chi_{max}=3.0$ and $\Omega_{0}=2.0$. It becomes evident by comparing the plots how despite the oscillations the cell concentrations decay significantly faster as we increase the dose and the action magnitude. The plot (**d**) results from the standard choice of drug parameters in Table 4 and outlines the tumor growth paradox, since the trajectory corresponding to the active therapy leads to the worst-case scenario of an out-of-control stem cell proliferation, with the activator a diverging and the CSC concentration reaching the upper bound, in contrast to the untreated case, where the tumor regresses in finite time after an overshoot. In this specific example, we take a large initial value of a, which in the critical case will diverge immediately, reason why it is not visible in the plot.

**Figure 6 cancers-15-00677-f006:**
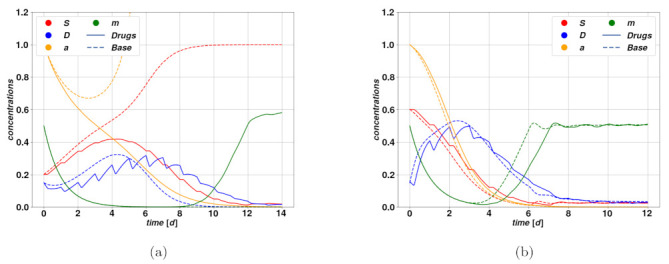
Trajectories overlay with the active cytostatic treatment case (solid lines) together with the untreated case (dashed lines). In all figures, we keep the same codes of colors as previously for *S*, *D*, a and m. In these examples, we assume the drug action to operate by reducing the mitotic factor λ. In (**a**) on the left the treatment is successful and prevents a dangerous proliferation associated with divergent stem cell activator a, and consequent population overtaking of CSCs over DCs. The trajectories in (**b**) on the right depict an overshoot case, where the untreated tumor undergoes autonomous regression, and the treatment is ineffective both in reducing the overshoot and enhancing the regression. The system and drug parameters are given in Table 2, Table 3 and Table 4.

**Figure 7 cancers-15-00677-f007:**
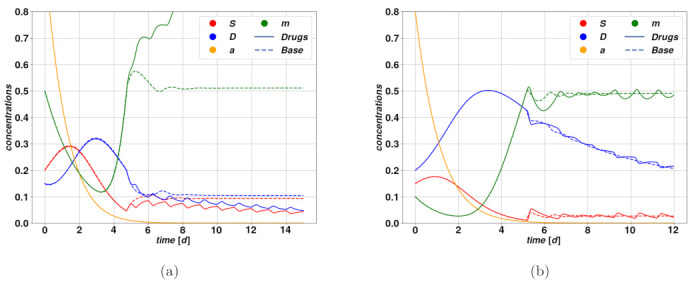
Trajectories of the system variables using a drug that inhibits the plasticity of the differentiated cells by reducing the plasticity factor $q_{0}$ that regulates the de-differentiation rate (solid lines). The trajectories corresponding to the untreated case are provided for comparison (dashed lines). We keep the same codes of colors as previously for *S*, *D*, a and m. (**a**) represents a case of effective treatment, in which the therapy is able to break the feedback loop associated with the plasticity mechanism, and lead the tumor to regression. In (**b**) the trajectories depict a case where treatment on the plasticity factor is not strong enough to lead to a regime change of the tumor going from a self-sustained and stable evolution to progressive annihilation as in (**a**). The relevant parameter values can be found in Table 2, Table 3 and Table 4.

**Figure 8 cancers-15-00677-f008:**
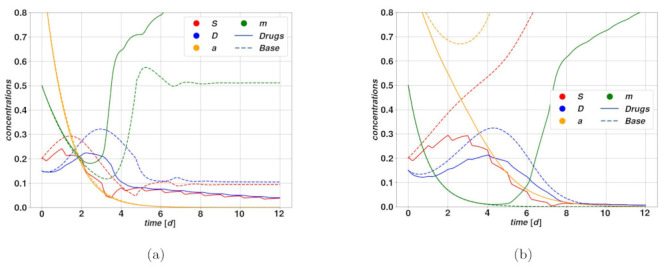
Trajectories of the system corresponding to the therapy combining a stimulated differentiation and a cytotoxic action (solid lines). The dashed trajectories correspond to the untreated case. We keep the same convention of colors as in the previous plots. Both plots (**a**,**b**) represent cases of effective treatments, under similar conditions as in Figure 7a and Figure 6a in order, respectively: in (**a**) the therapy breaks once again the feedback which produces the stable state associated with $F_{2}$, while in (**b**) the therapy is able to bring under control and to complete regression of a tumor that would otherwise dangerously diverge. One can notice that the net effect of enhanced differentiation and increased DC death rate is an induced nonzero death rate for the CSCs. In these examples, we assumed the actions $\Omega_{diff}$ and $\Omega_{cyto}$ to be equal in magnitude. The relevant parameter values can as usual be found in Table 2, Table 3 and Table 4.

**Table 1 cancers-15-00677-t001:** Dynamical variables of the basic model.

Name	Symbol	Order of Magnitude	Source
Cancer stem cells	S	1–5%	[27,28,29]
Differentiated cells	D	20–50%	[27,28,30]
Inert, healthy, quiescent cells	$N_{0}$	1−S−D	[27,28,30]
Activator	a	α/β∼0.1–0.5	[4,21,31,32]
Inhibitor	ψ	0.5–2	[29]
Plasticity coefficient	q(**m**)	$q_{0}∼$0.2–2	[33,34,35]

**Table 2 cancers-15-00677-t002:** Choice of parameters concerning biochemical signals.

Name	$\alpha_{a}$	$\alpha_{m}$	β	γ
Figure 1a	0.5	1.0	2.0	1.0
Figure 1b	1.0	1.0	2.0	1.0
Figure 2a	0.5	1.0	2.0	1.0
Figure 2b	1.0	1.0	2.0	1.0
Figure 2c	0.4	0.4	2.0	1.0
Figure 2d	1.0	1.0	2.0	1.0
Figure 3a	1.0	1.0	2.0	1.0
Figure 5a–c	1.0	1.0	2.0	4.0
Figure 5d	1.0	1.0	2.0	1.0
Figure 6a	0.7	1.0	4.0	1.0
Figure 6b	1.0	1.0	3.0	1.0
Figure 7a	1.0	0.5	2.0	3.0
Figure 7b	1.0	1.0	2.0	1.0
Figure 8a	1.0	0.5	2.0	3.0
Figure 8b	0.7	1.0	4.0	1.0

**Table 3 cancers-15-00677-t003:** Choice of parameters concerning cancerous cell populations.

Name	η	ψ	d	$q_{0}$
Figure 1a	5.0	1.0	1.2	0.5
Figure 1b	5.0	1.0	0.2	1.0
Figure 2a	5.0	1.0	1.2	0.5
Figure 2b	5.0	1.0	0.2	1.0
Figure 2c	2.0	1.0	0.4	1.0
Figure 2d	5.0	1.0	0.2	1.0
Figure 3a	5.0	1.0	0.2	1.0
Figure 5a–c	2.0	1.0	0.95	1.0
Figure 5d	1.75	1.0	0.8	1.0
Figure 6a	4.0	1.0	1.2	1.0
Figure 6b	1.0	1.0	0.8	1.0
Figure 7a	5.0	1.0	0.9	1.0
Figure 7b	5.0	1.0	0.2	1.0
Figure 8a	5.0	1.0	0.9	1.0
Figure 8b	4.0	1.0	1.2	1.0

**Table 4 cancers-15-00677-t004:** Choice of kinematic and dynamical parameters of the drug model.

$\chi_{\max}$	$\alpha_{d}$	*T*	τ	$\Omega_{0}$	$\chi_{0}$	$s_{\chi }$
1.0	$10.0\;d^{-1}$	1.0d	0.08d	1.0	0.2	0.05

## Data Availability

Not applicable.

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
