# Peer review of "Stochasticity and Drug Effects in Dynamical Model for Cancer Stem Cells"

_cancers, 2023, doi:10.3390/cancers15030677_

Round 1

Reviewer 1 Report

see the report

Author Response

see attached pdf

Reviewer 2 Report

This model evaluates a mathematical model of cancer stem cells and differentiated cancer cells. 

o  The results from the evaluations are specific to the particular model presented.  It may be difficult to generalize any results to real  biology.

o  It is unclear how equations (2) and (3) link back to equations (1).  This link is crucial to understand how treatment affects cell viability.

o  Evaluation of stochasticity in the system provides limited insight, given that the model itself is limited from a biological perspective.  In any case, this evaluation is not carried forward to the simulations of therapeutic response.

o  All the figure captions should describe the meaning of the variables in the figure insets.

o  There should be a table of variables with their names and meanings, and also a table of parameters, with their names, meanings, values, and sources.

o  The evaluation of plasticity should be expanded to better describe the biological/clinical implications.

Author Response

see attached pdf

Reviewer 3 Report

The article "The Stochasticity and drug effects in dynamical model for cancer stem cells" by L.Mori and M. B. Amar considers a kinetic model of heterogeneous cancer cell population mainly for investigating the conditions of  Cancer Stem Cell population stability and also to study the impact of stochasticity by introducing fluctuating factors and parameters. The aim is to decipher uncorrelated and heterogeneous clinical cancer drug treatment results. The article's added value is based on the equations' simplicity, where even for a simple dynamical case (model), the stability of the cancer cell population is a complex issue. Despite that, the article has a mathematical basis is well-written for the general audience, including clinical doctors, biophysicists and cancer biologists. Therefore, it is highly recommended for publication in Cancers.

Minor issues

1.       Because the article addresses the general audience, please explain the Wiener process briefly.

2.       It is expected for biological systems to have memory (non-Markovian). However, the model describes the dynamic evolution of memoryless systems (Markovian). It should be understood that the mathematical description of a dynamical system with memory has bifurcating points and attractors, which are difficult to identify. Also, they are highly unstable to small perturbations, leading to possible chaotic behaviour. However, this point does not limit the merit of the article. On the contrary, it emphasises the importance of the work. Please, insert a small paragraph at this point (page 7 line 267).

3.       Please, explain the meaning of plasticity parameter m.

4.       The model does not include a diffusion term commonly used in such models. Please explain.

5.       There is room for model description improvement. Please explain better the parameters α, m and Nint.

Reviewer 4 Report

Mathematical models delineating the dynamic changes of the CSC populations with and without drug treatment are crucial in understanding the mechanism of CSC propagation and in predicting therapeutic efficacy of certain drugs. This work made commendable attempt in this direction. However, a major flaw in this paper is that it assumes CSCs have high proliferation rate and grow fast based on reference 17. In fact, ample evidence from various models indicates that CSCs maintain themselves in a quiescent state (called dormancy) that mainly account for drug resistance (e.g., PMID: 35505363, DOI: 10.1186/s13287-022-02856-6). Therefore, the foundation of this paper remains flawed. Furthermore, according to the authors, there are two types of drugs targeting CSCs, namely cytotoxic and cytostatic drugs. In fact, there is another type of drugs that can convert the CSCs into differentiated cancer cells. When combined with traditional chemotherapy or targeted therapy, this type of anti-CSC drugs has the potential of eliminating all cancer cells. This should be added into the current model.

Round 2

Reviewer 1 Report

Thanks for addressing all the comments. I suggest for publication.

Reviewer 2 Report

Thank you for the updates to the manuscript.

Reviewer 4 Report

The authors responded to my comments in an appropriate manner, and the revised manuscript is now acceptable for publication.